# Consideration of Power Transmission Characteristics in a Micro-Gear Train

**DOI:** 10.3390/mi15020284

**Published:** 2024-02-17

**Authors:** Yasuhiko Arai

**Affiliations:** Department of Mechanical Engineering, Faculty of Engineering Science, Kansai University, 3-3-35, Yamate-cho, Suita 564-8680, Osaka, Japan; arai@kansai-u.ac.jp

**Keywords:** micro-gear train, loss of transmitted torque, 3D-printed micro-gear, differences between micro- and macro-gear trains, mathematical and statistical processing

## Abstract

Characteristics related to power transmission in the micro-domain, based on dry rolling contact of the gears, were investigated using a 3D-printed gear train with a pitch circle diameter of 84 µm in order to experimentally compare the power transmission efficiency in the macro- and micro-domains. For a basic gear train with two intermeshing gears, it was shown that the gear train in the micro-domain was capable of transmitting power to the same extent as in the macro-domain. However, in gear trains with complex power transmission paths, assuming a planetary gear train with multiple meshing gears, it has been shown that the power transmission characteristics of micro-domain gears differ from those in the macro-domain. The use of gear trains in the micro-region necessitates consideration of the loss of transmitted torque due to contact between tooth surfaces, which is unique to the micro-region and different from its use in the macro-region.

## 1. Introduction

MEMS (Micro Electro Mechanical Systems)-based fabrication has been developed as an integrated device technology for various functions in various fields, such as machinery, electronics, optics, and chemistry [1,2,3]. In the semiconductor field, various sensors, micro-motors, and other actuators have been produced based on Si microfabrication production technology [4,5,6,7]. However, when a rotating shaft comes into contact with a shaft wall surface, such as in the case of micro-motors, the problem of structural damage due to adhesion wear is raised. To address this problem, microstructure fabrication techniques using materials other than Si are being developed [8,9,10].

Peripheral technologies related to power transmission have also been reported, such as the development of optical bearings using optical radiation pressure [11], in an attempt to reduce the loss of energy efficiency due to the generation of friction between the shaft and shaft wall between microstructure surfaces. Another type of peripheral technology involves the design and fabrication of micro-scale gear trains for effective torque transmission and utilisation [12,13,14,15,16].

However, it is well-known in mechanical engineering that within involute gears, which are commonly used in power transmission mechanisms, there is rolling contact between tooth surfaces during power transmission, as well as slippage between tooth surfaces [17]. In other words, meshing losses, as frictional losses on the tooth surfaces, reduce power transmission characteristics [17].

In the macro-domain, oil lubrication is generally used to address these phenomena. [17] However, in the micro-domain, fluid lubrication is thought to significantly degrade the motion characteristics. For example, in micro-area motions, such as in MEMS, even if the surrounding fluid is air, the Reynolds number of the fluid becomes small, and dust, once lifted, does not fall easily. Thus, in everyday life, micro-objects in a fluid can be observed to exhibit behaviour different from the mass-driven motion in the macro-range. Therefore, when gears are used in the micro-range, they must be treated as mechanical elements using a power transmission mechanism with dry rolling contact phenomena to achieve their original power transmission function.

There have been many reports regarding the material properties, manufacturing methods, and service life of micro-gears alone as mechanical elements [12,13,14,15,16]; however, the power transmission characteristics of micro-gears in a gear train have not been sufficiently considered. It is conceivable that in the micro-range, as mentioned above, surface-to-surface contact may cause a greater loss of transmitted power than in the macro-range. Therefore, in a power transmission mechanism using micro-gears, which involves sliding in addition to rolling contact, a significant loss of energy is likely to occur if the gear train is manufactured carelessly. Given that these power transmission characteristics have not been clarified, it is difficult to design a micro-gear train for effective power transmission considering the loss of transmitted torque. This study investigates the power transmission characteristics of a micro-gear train, which are considered essential for designing a micro-gear train for actual power transmission as an MEMS device.

In this study, characteristics related to power transmission by dry rolling contact were experimentally compared and examined in terms of the transmission efficiency of gears in the macro- and micro-ranges; moreover, I examined the differences in characteristics between the power transmission mechanism of micro-gears in the micro-range and that of gears in the macro-range. The results of this study are used to discuss the type of gear structure and phenomena that need to be considered when using gears in the micro-range.

Specifically, as an example, this study examined the power transmission characteristics of an involute gear with a pitch circle diameter of 84 µm, a module of 6 μm, 14 teeth, and a pressure angle of 20 degrees, whose tooth surface was metallised by sputtering aluminium on the surface of a micro-spur gear fabricated using resist with a micro-3D printer [11,18], capable of microfabrication through two-photon absorption.

Furthermore, when investigating the power transmission efficiency, it is believed that, by using an inspection method based on the same measurement principle to evaluate power transmission efficiency in the macro- and micro-domains, it is possible to directly compare the characteristics of the gears in both domains. The moment balance method has been reported as one of the methods for measuring power transmission characteristics [19]. In that study, a torque detection mechanism using an ultrafine platinum wire was constructed such that the moment balance method [19] could be used in the micro-domain, and it was used to evaluate the power transmission efficiency of gears in the micro-domain.

The results of this study show that, for a simple two-meshing-gear train, the dry rolling contact power transmission efficiency in the micro-range can transmit power to the same extent as in the macro-range. However, when a planetary gear train with several meshing gears was considered, and the loads applied to the gear train were taken into account, the power transmission characteristics of the micro- and macro-gears in a composite gear train were confirmed to be different.

We believe that the results of this study will encourage further use of power transmission mechanisms with micro-gears in the future.

## 2. Materials and Methods

### 2.1. Fabrication of Micro-Gears Using a 3D Optical Fabrication System

In this study, a gear train was fabricated as an inspection target using a Nanoscribe Photonic Professional GT, which can easily produce microstructures with complex shapes. Using this machine, a gear train with a three-dimensional resolution of 200 nm and extremely high manufacturing accuracy can be produced in a single manufacturing process. However, because the size of the object is limited to a fabrication space of 300 × 300 × 300 μm^3^, the gear train used in this study is a spur gear, which can be relatively easily fabricated within this fabrication size (module: 6 μm, pitch diameter: 84 μm, tooth thickness: 20 μm, number of teeth: 14, pressure angle: 20 degrees). This spur gear was used as a test piece for the experiments. As the gears produced by the machine were made from a polymeric material with a solidified resist (IP-dip), a metallic tooth surface was formed by sputtering a 200 nm thick metallic film of aluminium onto the gear surface. Visual inspection of the SEM images of the gear after manufacturing confirmed that the aluminium had been deposited on the tooth surfaces. As a result, the tooth surface would make contact as a metallic surface when the gears were engaged.

This allowed for the effect of the rolling contact of the teeth on the transmitted torque to be considered as a contact phenomenon between metal surfaces, and the structure of the gear teeth to be produced mimicked that of the exoskeleton of a crustacean, such as the shell of a crab. As a result, the experiment was performed using gears that were expected to increase the strength of the structure and tooth surface.

An example of the gears used in the experiment is shown in Figure 1. This is the simplest gear train and features two intermeshing gears. Figure 1a shows the corresponding 3D CAD design drawing. In this gear train, to prevent the two gears from fusing together during manufacturing, they were designed using a structure that is manufactured in stages. The first stage of manufacturing does not include meshing [18], as each of the two gears is supported at three points and are later meshed by removing the supports.

Figure 1b shows an SEM image of the two gears meshed together by sputtering aluminium and removing the support after fabrication via the 3D printer.

The SEM image confirms that the gears were fabricated according to the design shown in Figure 1a. Thus, in this study, the micro-3D printer was used to produce the gear train exactly according to the design.

### 2.2. Measuring Power Transmission Efficiency Using the Moment Balance Method

#### 2.2.1. Moment Balance Method

Figure 2 shows an overview of how the moment balance method [19] is used to determine power transmission efficiency. This method is used to measure the power transmission efficiency of gears by applying a quasi-static torque to the gear.

As shown in Figure 2, gears with an equal pitch circle diameter (D) are meshed and a gradually increasing load W1 is applied to the input side to generate a torque of T1 = D/2 × W1 on the input gear. The load W2 is measured at the moment when the torque balance is lost, and the torque on the output side is determined as T2 = D/2 × W2. The results of the torques applied to each gear are then used to determine the power transmission efficiency as T2/T1.

The moment balance method is used to determine power transmission efficiency by examining the torque balance between the input and output gears. For example, if the torque applied to the input shaft is T, the torque obtained from the output shaft can be determined by multiplying the input torque transmission efficiency as T × η.

However, if the torque T applied to the input shaft is less than the minimum torque required to rotate the output shaft, then no rotational torque can be obtained from the output shaft. Furthermore, if, for example, a torque of 2T is applied to the input shaft and the minimum input torque at which the output shaft can rotate is Tin_min, then the output shaft torque will not be 2T × η by simple multiplication; instead, it will be related to 2T − Tin_min. Note that this phenomenon occurs in the transmission torque.

#### 2.2.2. Detecting the Power Transmission Characteristics of Macro-Gears Using the Moment Balance Method

Before investigating the power transmission efficiency of gears in the micro-range using the moment balance method, the power transmission efficiency of gears in the macro-range was measured using the moment balance method to verify the possibility of using this method in the micro-range. The experimental setup is shown in Figure 3.

This study used gears (manufactured by Kyouiku Gears MFG, Co., Ltd. Tokyo, Japan, model number: S1D120b-0608) made of a polymer material (polyacetal) with a relatively large pitch circle diameter, which makes them easy to handle, and a smooth involute curved surface, which is believed to provide smooth rolling contact and smooth gear rotation.

Specifically, the measuring principle of the moment balance method and the power transmission efficiency based on the results were investigated when the two gears were meshed (module: 1, pitch circle diameter: 120 mm, number of teeth: 120, tooth thickness: 6 mm, pressure angle: 20 degrees) based on the following specifications.

The instrument based on the moment balance method, shown in Figure 3, supported the shafts of the two gears with bearings. The centres of rotation of the two input and output gears were aligned using the three-dimensional stage (3D stage A) shown in Figure 3 and precisely meshed on the pitch circle of the gears. In addition, a 0.200 kg weight mounted on the electron balance was suspended by a flexible wire around the circumference of the pulley attached to the gears, as shown in Figure 2. The weights of the two gears reset the initial reading of the electron balance, and the balance of the torque on each gear was set at the start of the experiment by adjusting the height of each three-dimensional stage so that the output was zero in this state.

After the initial setup, when the forces were balanced in the device, the output of the electronic balance was continuously recorded on a computer, and a video was recorded. This created a measurement system that could accurately record the load applied when the gear on the input side began to move.

With this setup, when the input-side three-dimensional stage (3D stage B) of the electronic balance was gradually lowered in the direction of gravity, as indicated by the red arrow, the memory of the output-side electronic balance changed abruptly. By reading the values of the input-side electron balance and the stored results of the output-side electron balance, the power transmission characteristics were measured based on the torque applied to each gear at the moment of imbalance.

The results of this experiment, which was performed 20 times, showed that the average value (Ave-macro) was 94.3% (sum of squares 68.2, unbiased variance 3.4). This average value was found to be in good agreement with that published by the gear manufacturers [20,21]. These results suggest that the power transmission efficiency of spur gears can be measured by the moment balance method, as previously reported [19], so this method was used to measure the power transmission characteristics.

#### 2.2.3. Micro-Domain Measurement of Power Transmission Characteristics Using the Moment Balance Method

To apply the moment balance method used to measure the power transmission characteristics of gears in the macro-domain to the micro-domain, an apparatus was set up to measure the power transmission characteristics, as shown in Figure 4b, according to the conceptual diagram shown in Figure 4a.

The measuring principle was based on that of the device used to measure the power transmission characteristics of the macro-model gear shown in Figure 3, as explained in Figure 4a.

A force is applied to the input gear in the tangential direction of the pitch circle using a platinum wire (diameter 1 μm) and a piezoelectric element. At this time, the input gear remains continuously stationary, so that the platinum wire on the input side gradually deflects as the tangential displacement by the piezoelectric element increases. When a certain force is applied, the output gear rotates. This causes the platinum wires on the output side to deflect. The extent of the deflection of the ultrafine platinum wire is measured from the recorded video. The micro torque wrench end was able to carry out the experiment without slipping on the gear tooth surface because it was firmly attached to the tooth.

The forces acting on the input and output platinum wires are calculated from the measured deflections using the finite element method (COMSOL multiphysics), as shown in Figure 5. The torque is then calculated as the product of the length of half the pitch diameter of the gear and the force determined by the finite element method. The power transmission torque in the micro-range is obtained using the ratio of moments between the input and output, as in the moment balance method in the macro-range.

Specifically, as shown in Figure 4b, the base of the intermeshing gears produced by a 3D printer was fixed with adhesive, and the tip of an ultrafine platinum wire with a diameter of 1 µm was placed on the pitch circle of each gear from both sides with respect to the intermeshing gears. A mechanism was then constructed so that the platinum wire on the input side moved 10 µm parallel to the tangential direction of the pitch circle of the gear on the input side using a piezoelectric element. A force was applied to the input side, and the input-side torque was determined from the extent of deflection of the ultrafine wire as the input-side gear rotated.

In contrast, the ultrafine platinum wire attached to the gear on the output side was fixed, the gear on the output side rotated when the gear on the input side began to rotate, and the rotational torque caused the ultrafine platinum wire on the output side to deflect. The extent of this deflection was recorded as numerical data by using a 20× objective lens and using the pixel size as a reference for the deformed ultrafine wire, as one pixel of this camera was 0.387 µm wide. Through this process, the state of equilibrium between the torque applied to the gear on the input side and that on the output side was detected as a change in the deflection of the ultrafine wire on the input and output sides.

Specifically, the moment when the ultrafine platinum wires from the left and right, shown in Figure 6a, were placed on the pitch circle of the micro-gear and started to push the teeth, as indicated by the white arrows, and the rotating gear shown in Figure 6b was recorded as an animation under a microscope. Moreover, the gear movement of the input and output was recorded as the extent of deflection of the ultrafine platinum wires.

The results were also used to determine the forces acting on the microcantilever, based on the finite element method, and to obtain the rotational torque for the input and output. The ultrafine platinum wires used for torque detection were silver-coated and used as the force-sensing element of the microcantilever by removing the silver with nitric acid. For this reason, measurements in the micro-torque range were also carried out with several different platinum wires, although it was not always possible to obtain straight and constant lengths of the rod-shaped platinum wires. The force measurements were also performed using a slightly bent ultrafine wire. In such cases, even if the measurement results were bent, they could be performed without any variation because of the finite element method processing, which accurately captured the shape of the platinum wire. The power transmission efficiency could be determined by observing the balance of moments between the input and output during this deformation.

## 3. Results and Discussion

### 3.1. Investigation of the Power Transmission Characteristics of Two Micro-Spur Gears

Table 1 presents an example of the procedure for measuring the power transmission efficiency of the micro-gears shown in Figure 6. As shown in Table 1, the deflection of the ultrafine platinum wires on the input side (Gear-1) and output side (Gear-2) were measured under a microscope as 11.43 μm and 8.78 μm, respectively, based on the pixel size of the camera, as described above. From these results, the forces acting on the ultrafine platinum wires on the input and output sides were determined to be 3.06 μN and 2.87 μN, respectively, using the finite element analysis software COMSOL Multiphysics Version 6.1. In this measurement, the forces acting on the gear did not differ significantly according to the finite element processing, although different deformations were obtained with different lengths of the microcantilever (depending on the length of the platinum wire). The obtained forces were used to calculate the torque.

Given that the pitch circle diameter of the micro-gear was 84 μm, and assuming that each of the above forces acted on the pitch circle of a gear with a radius of 42 µm, the torque acting on each gear was estimated to be 0.129 pNm on the input side and 0.121 pNm on the output side. A power transmission efficiency of 93.7% was determined from the input-to-output ratio.

The same experiment was repeated 10 times by changing the microcantilever. The results showed that the average power transmission efficiency of the micro-gear was 94.1%, as shown in Micro-Gear Train in Table 2.

By comparing these results with the power transmission efficiency of the gear train in the macro-region (Macro-gear train in Table 2) obtained from the experimental apparatus shown in Figure 3, the differences in gear characteristics between the micro- and macro-regions were investigated, and it was determined whether the design guidelines for gear trains in the macro-region, which have traditionally been studied in detail, could be used in the micro-region.

A mathematical-statistical [22] comparison of the results of the two measurements, micro and macro, was conducted to investigate whether the above design of the gear train in the micro-range could be realised based on the idea in the macro-range. The first step was to determine whether the two datasets were normally distributed. Based on the results, an analysis of variance [22] and a test for the difference of means [23,24] were then performed between the two sets of data to compare the characteristics of the micro-domain gears with those of the macro-domain gears.

The results of the graphical test of the Gaussian distribution of the respective measurement results in the macro- and micro-regions are shown in Figure 7. As the cumulative distribution function varies linearly in both the micro- and macro-regions, the respective measurement results are considered to be normally distributed [25]. In this study, the Anderson–Darling test [26] was also performed to quantify that they are normally distributed. The resulting *p*-values in the Anderson–Darling test for the micro- and macro-gears were all above 0.05, as listed in Table 2. From this result, at the 95% confidence level, the results of the Anderson–Darling test, together with the results of the cumulative distribution function graph, indicate that the measurement results of the micro- and macro-gears are normally distributed.

Based on these results, the variance of the micro- and macro-gears was then tested [22,23] and, as shown in Table 2, using the F distribution, the degrees of freedom were set to nine and nineteen in the respective data. The ratio of unbiased variance showed that F0 = 2.71 < 2.89 = F(9, 19; 0.025), meaning that it could be determined with 95% confidence that there was no difference in variance between the two groups.

Furthermore, based on the results, a test for significant differences in the differences between the means of the respective micro- and macro-data [23,24] was performed using the t-distribution. As shown in Table 2, from *t*0 = 0.00223 < 2.048 = *t*(28, 0.025), no difference was found between the means of the two groups at the 95% confidence level.

From the above results, it is considered that the effects of rolling contact and slippage on the tooth surfaces, which are the most important factors in the power transmission characteristics of micro-gears, are only affected to the same extent as those of macro-gears for gears with a pitch circle diameter of approximately 100 µm. It was therefore considered that a micro-gear train in which two gears with a pitch diameter of approximately 100 µm mesh could be designed using the same approach as the design guidelines for macro-gears.

### 3.2. Investigation of the Power Transmission Characteristics of a Gear Train with Four Micro-Spur Gears under Load

#### 3.2.1. Gear Train with Four Micro-Spur Gears Used in the Experiment

As shown in the previous sections, the power transmission efficiency of the two micro-gears without load was found to behave in the same manner as the power transmission efficiency of gears in the macro-range. However, when a gearbox is used, a load exists. Therefore, the next step was to examine the power transmission characteristics of the micro-range gear train when a load was present, considering the gear train with a more complex arrangement of multiple meshed gears, such as planetary gears.

When designing a normal gear train, there are two types of systems: one transmits power from a single gear to several gears via a rotating shaft, and the other transmits power indirectly by meshing several gears with a central gear, such as planetary gears [17]. In this study, the meshing structure of the gear train shown in Figure 8a, whose power transmission characteristics could easily be compared at the macro- and micro-levels, was investigated. The gear train has characteristics that account for the properties of both a system that transmits power directly to multiple gears (in which power transmission is considered to generate a load on the input gears) and a system that transmits power via intermediate gears such as planetary gears (in which power transmission is considered to generate a load on the input gears owing to the complex meshing of the gears). By studying this gear train, the power transmission characteristics of the gears in the micro-region were investigated when a load was generated on the gears.

However, by setting the number of surrounding gears (Gears-A, -C, and -D) to three, as shown in Figure 8a, the power transmission characteristics considering the load for the case where several gears meshed with Gear-B, installed in the middle, were considered possible.

In this study, when the pitch diameter differs between the central gear and the surrounding gears, the torque used for measuring the power transmission characteristics can be converted during the torque calculation by using the radius of the gears as a parameter. However, to avoid using more parameters than necessary during the evaluation, it was decided to study the power transmission characteristics by combining gears of the same size throughout, as far as possible.

According to this concept, a gear (Gear-B) was installed in the centre, as shown in Figure 8b, as a gear train that could be produced within the production range (300 μm × 300 μm) with the 3D printer used in this study. Three identical gears (Gear-A, -C, and -D) were then installed around this gear.

Thus, gears of the same size as those used in the case where the meshing characteristics of the two gears shown in the previous section (module: 6 μm, pitch circle diameter: 84 μm, tooth thickness: 20 μm, number of teeth: 14, pressure angle: 20 degrees) were used to study the power transmission characteristics in the case of multiple meshing.

During the experiment, the power transmission characteristics were studied by carrying out the rotational torque of each gear, as shown in Figure 8a, in the same manner as in the process based on the moment balance method with two gears described in the previous section. In other words, when studying the power transmission characteristics of four gears in the micro-region, as shown in Figure 8b, the power transmission characteristics according to the moment balance method were also studied when the four gears in the macro-region were meshed, as shown in Figure 8a, and the differences with the micro-region were studied using the apparatus shown in Figure 9.

#### 3.2.2. Power Transmission Efficiency of a Gear Train with Four Gears Considering the Load

Experiments on power transmission from a peripheral gear (Gear-A) through Gear-B to the two surrounding gears (Gear-C and -D), shown in Figure 8a, were carried out using a micro-torque wrench made of platinum wire on the input gear, as shown in Figure 6a,b, to measure the torque, as shown in Figure 10a. 

In this case, a rotational torque was applied by using a power rod to push and rotate the gear. The torque was measured from the deflection of the platinum wire. By detecting the balance of the torque, the respective rotational torques of Gears-C and -D were measured as Gear-A rotated, as shown in Figure 10b,c. The power transmission efficiency was then measured from the results.

The resulting calculation process for determining the power transmission efficiency is shown in Table 3 as an example. In this case, as in the calculation process shown in Table 1, the deflection of the platinum wire was measured from the video, and the finite element method was used to determine the force acting on the gear from the deflection of the platinum wire at Gears-A, -C, and -D. The torque was determined based on this force. The ratio to the torque at Gear-A, as the input gear, was then determined as the power transmission efficiency. In the example shown in Table 3, the torque of 432.0 pNm, given as input to Gear-A, was found to be transmitted as 161.0 pNm in Gear-C and 153.0 pNm in Gear-D. From these measured torques, it can be seen that the power transmission efficiencies for both Gear-C and Gear-D are approximately 35%.

However, measurement methods based on the moment-balance method have been found to have certain problems. When detecting the power transmission characteristics using this type of method, the torque occurring in the gears is taken as a change in balance with the input torque and is detected by the ratio of the two. Therefore, the torque in the gears must be measured based on the difference with the input torque. Additionally, if, as in the present case, the aim is to measure the torque applied to Gear-B at the centre of the gear train, as shown in Figure 8a, a certain amount of torque must be applied to Gear-B beforehand. When such an additional torque is applied, the test conditions are different from those of the original procedure for measuring the transmission torque from Gear-A to Gear-C and -D.

For this reason, the torque of Gear-B was deliberately not measured in the experiments conducted in this study. Thus, the results of this study were based only on the results of the power transmission characteristics between the input and the two outputs.

In the gear train shown in Figure 8a, Gear-D was considered as the load transmitting power from Gear-A to Gear-C, and Gear-C was considered as the load transmitting power from Gear-A to Gear-D. In other words, in a gear train with the structure shown in Figure 8a, the power transmission characteristics of a gear train with a load and a structure in which multiple gears are meshed to distribute the torque from one gear to two gears were measured. Several phenomena can be considered when examining such a gear train.

In addition, actual measurements have shown that, in some cases, the power transmission characteristic results for Gear-C and -D in Table 3 are not always the same as the torque halved from Gear-B. This is attributed to problems with the meshing of the two gears. 

In such a case, if the power transmission characteristics of the individual gears of Gear-C and -D were to be studied, complex considerations would be required, such as solving the problem of handling mismatched transmission torques for each gear. Therefore, in this study, because the original purpose was to examine the power transmission characteristics in the micro-region, the relationship in which the values of Gear-C and -D differ and vary is not analysed in detail; instead, the overall relationship from Gear-A as the input to Gear-C and -D as the output is considered. Therefore, I decided to examine how the power was transmitted. In other words, by taking the sum of Gear-C and Gear-D as the total output torque, the ratio between Gear-A and Gear-C + Gear-D was treated as the total power transmission characteristic, as shown in Table 3. As an example of the measurement results shown in Table 3, a transmission characteristic of 72.7% was treated as the total power transmission characteristic.

#### 3.2.3. Comparison of the Power Transmission Characteristics of Four Micro-Spur Gears with Those of a Gear Train in the Macro-Domain

By repeating this measurement 10 times, as in the case of the two gears, it was found that the average value of the power transmission efficiency of the micro-gears was 69.6%, as shown in Micro-gear train in Table 4.

In the results up to the previous chapter, the power transmission efficiency between the two gears shown in Figure 6 was approximately 94%, as shown in Table 2, and it was concluded that a transmission efficiency of approximately 100 µm can be achieved, which is no different from the results for the macro-range. However, if a slightly more complex power transmission path is established in the gear train, and the respective gears are loaded on top of each other, the transmission characteristics decrease to approximately 70%, as shown in Table 3.

As in the case of the study of the differences in the transmission characteristics of two gears, the differences in the transmission characteristics of the gears in the micro- and macro-regions were also studied in this case, following a comparison with the results for the gear train in the macro-region, as shown in Figure 9.

In the case of the four-gear macro-regional gear train shown in Figure 9, the experimental setup was constructed according to the schematic diagram of the gear train shown in Figure 8a, using four of the same gears as in the gear train shown in Figure 3 (module: 1, pitch circle diameter: 120 mm, number of teeth: 120, tooth thickness: 6 mm, pressure angle: 20 degrees) so that the same considerations could be made as when studying the transmission characteristics with two gears, as shown in Figure 3.

In Figure 9, Gear-A is supported on a 0.300 kg weight mounted on a digital electronic balance on Stage-A by a soft wire placed around the circumference of Pulley A (100 mm diameter), which is connected to Gear-A by a rotating shaft supported by bearings, as shown in Figure 8a. 

Gears-C and -D are also supported, as in the case of Gear-A, by soft wires placed around the circumferences of the pulleys (100 mm diameter) mounted on shafts supported by bearings on the respective gears (100 mm diameter), as shown in Figure 8b, and by 0.300 kg weights mounted on the electronic balance on Stage-C and -D.

The four-gear transmission apparatus was constructed with the same structure as that of the micro-model transmission shown in Figure 8b for the macro-model shown in Figure 9.

This apparatus was used to investigate the power transmission characteristics in the macro-range by recording the changes in the electronic balance connected to the output gear when the stage of the input gear (Gear-A) was gradually lowered in the direction of gravity, as in the experiment with two gears using the apparatus shown in Figure 3.

This experiment was performed 20 times in the same manner as that used for the two gears. Again, the ratio of the sum of the torques of Gear-C and -D to the torque of Gear-A was taken as the power transmission characteristic.

The results are listed in Table 4 Macro-Gear Train. The average power transmission characteristic in this case was 75.5%. The power transmission characteristics in the macro-range are also reduced, considering that the two gears had a transmission characteristic of approximately 94%.

However, the average value of the power transmission characteristics through the two gears did not differ significantly between the micro- and macro-regions and could be treated as having no statistically significant difference. However, when the gear train becomes slightly more complex, as shown in Figure 8a, and a load is generated between the input and output, the average values of the transmission characteristics in the micro- and macro-regions are different, being 69.6% and 75.5%, respectively.

Therefore, the transmission characteristics in the micro- and macro-regions were statistically processed in the same manner as those of the two gears.

The first step was to determine whether the measurement results were normally distributed. A graph of the cumulative distribution function of the measurement results obtained from each of the macro- and micro-gears is shown in Figure 11. Both measurement results are considered normally distributed because they vary linearly [25]. Furthermore, the Anderson–Darling test [26] showed that the two measurements were normally distributed at the 95% confidence level, as the *p*-value in the Anderson–Darling test was greater than 0.05, as shown in Table 4.

Assuming that the results were normally distributed, the micro- and macro-results were then tested for differences in dispersion using the F-distribution. The results showed that F0 = 1.146 < F(19.9: 0.025) = 2.88, indicating no difference in variance between the results of the two measurements. However, the results of the investigation using the t-distribution for the difference between the mean values of the two power transmission efficiencies, *t*0 = 5.56 > *t*(28, 0.025) = 2.048, showed a difference between the results of the mean values of the two.

The results suggest that the power transmission characteristics of the gears in the micro-region can be reduced in the presence of loads compared to the power transmission characteristics in the macro-region. These results suggest that the use of gears in the micro-region is more affected by the loss of power transmission due to contact (which is specific to the micro-region) than in the macro-region.

The transmission characteristics of the micro- and macro-gears, which were found to have similar transmission characteristics as in the case with two gears, were also investigated in a gear train with four planetary gears. It was found that the transmission characteristics between the micro- and macro-ranges in the four-gear train changed when a load was applied, even in experiments with the same gears, and it was confirmed that the power transmission characteristics did not change with two gears. It is believed that this phenomenon confirms the reduction in power transmission efficiency due to rolling contact, which has been a concern since the beginning of the use of micro-range gears.

Based on the above results, the following points should be considered when using a micro-domain gear train.

(1)First, it is necessary to consider the design of the gear train, taking into account the fact that the transmission loss in the micro-range is greater than that in the macro-range.(2)When performing activities using power in the micro-range, the mechanism should be designed so that the power source is installed as close as possible to the actual object to be driven, while the installation location of the power source should be considered so that transmission losses do not occur. If a gear train is used, the power transmission mechanism should be designed to use a simple gear train in which power is transmitted through a shaft.(3)In the micro-range, a gear train, such as a planetary gear train in which multiple gears mesh with a single gear, which may cause mutual loading, is not considered an effective design for reducing transmission losses.(4)If a power transmission mechanism using a gear train is to be used in the micro-range, it is necessary to consider a solid lubrication mechanism to replace oil lubrication in the macro-range [17].(5)If a power transmission mechanism using a gear train is to be used in the micro-range, the involute gears that are currently primarily used in the macro-range have convex tooth surfaces that come into contact with each other, resulting in a situation where a large frictional force acts on the gears. Therefore, it is believed that a detailed study of a gear train suitable for use in the micro-range is required, such as a module suitable for the micro-range, pressure angle, solid lubrication (as mentioned above), and increased tooth surface strength.

I consider that the results of the investigation of the power transmission characteristics based on the moment balance method used in this study can only be used to a limited extent, as the power transmission coefficient is obtained in relation to the static frictional force on the shafts of the gear train. However, for objects such as MEMS, where friction between the contacting surfaces of the structure is a problem, consideration of static friction is strongly required. Therefore, the moment-balance method is considered suitable for studying the power transmission characteristics of MEMS.

## 4. Conclusions

In the micro-region, where the friction between contacting surfaces has a significant influence on the kinematic function when structures come into contact with each other, a gear train that transmits power through rolling contact between tooth surfaces is expected to experience losses in power transmission efficiency. Therefore, when a gear train is used in a MEMS, the power transmission efficiency is expected to be significantly reduced. As a result, it is believed that the use of gear trains in MEMS needs to be approached according to a policy different from the traditional gear design approach used in the general macro-domain. In order to discuss this problem, a gear with a diameter of 84 µm was manufactured using a 3D printer and measured. In the course of the study, a measurement technique based on the principle of the moment balance method, which can be used in the micro-range, was proposed to measure the power transmission efficiency in the micro-range. Using this technique to observe the power transmission efficiency in the micro-range, a comparison was made between the power transmission efficiency in the micro-range and that in the macro-range based on the same measurement principle as that in the macro-range.

The results show that the power transmission efficiency between two unloaded gears in the micro-range is the same as that in the macro-range. In addition, to design a compact micro-gearbox in the future, a gearbox capable of splitting the torque from a single axis, assuming planetary gears, was also investigated. However, the micro-gears used in the study, which were expected to achieve this goal, showed a significant reduction in power transmission efficiency compared to the power transmission efficiency of the macro-gears. The results show that, when designing a power transmission mechanism using 100-μm scale gears in the micro-domain, it is necessary to consider design guidelines suitable for power transmission by gears in the micro-domain in addition to traditional gear design techniques developed in the macro-domain.

In the future, it is hoped that new related technologies, considering the design and manufacturing methods of power transmission mechanisms with gears in the MEMS domain based on traditional gear design techniques in mechanical engineering, will be developed to promote the use of gears in the micro-domain based on the results of this study.

## Figures and Tables

**Figure 1 micromachines-15-00284-f001:**
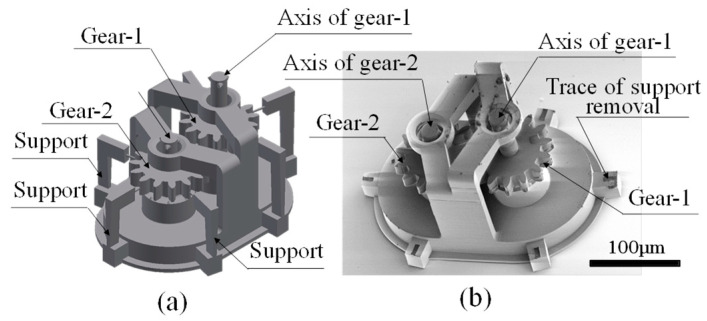
SEM image of micro-gear. (**a**) 3D CAD design. (**b**) SEM image of produced gear.

**Figure 2 micromachines-15-00284-f002:**
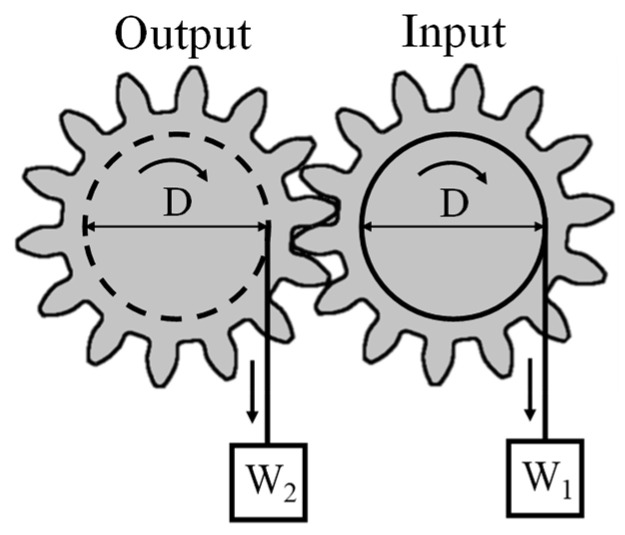
Principle of moment balance method.

**Figure 3 micromachines-15-00284-f003:**
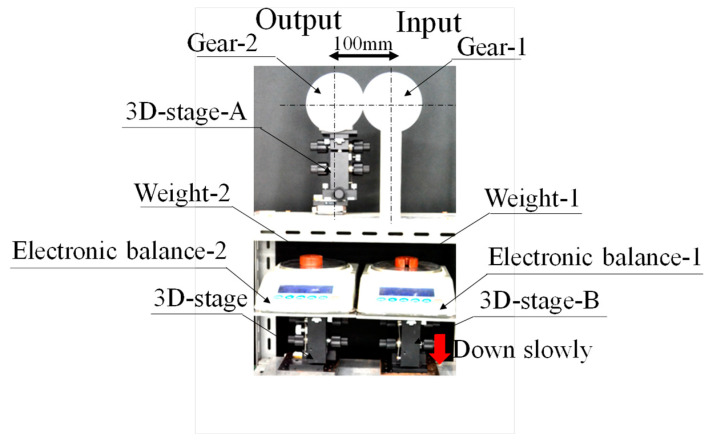
Equipment of measuring power transmission efficiency of macro gears by moment balance method.

**Figure 4 micromachines-15-00284-f004:**
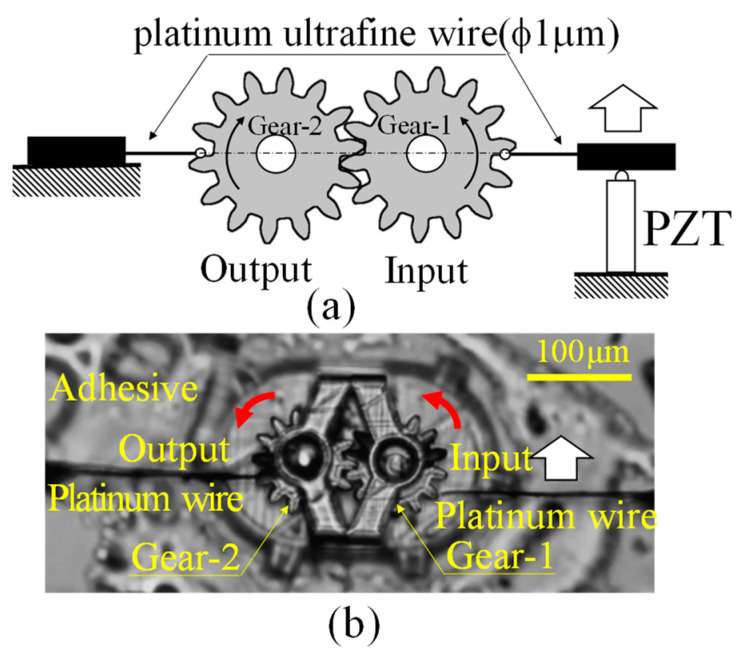
Moment balance method for micro gears. (**a**) Schematic of the method. (**b**) Real view of measurement.

**Figure 5 micromachines-15-00284-f005:**
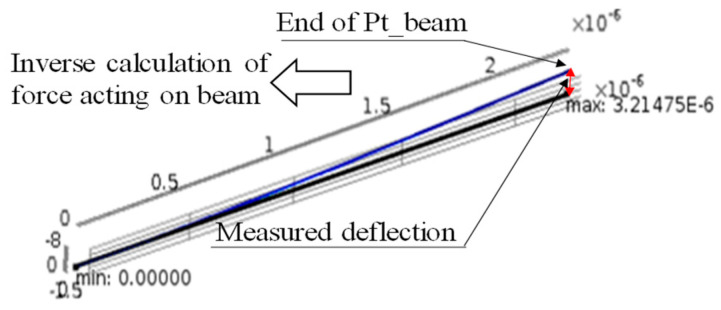
Calculation of forces using the finite element method.

**Figure 6 micromachines-15-00284-f006:**
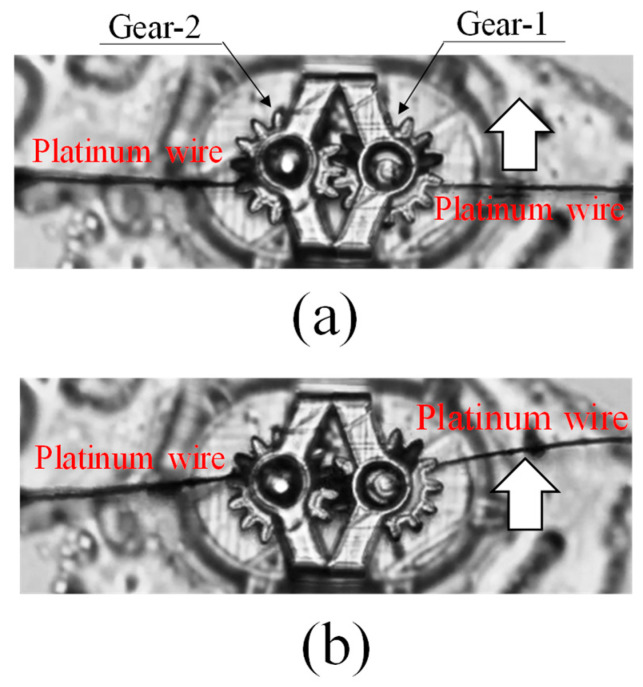
Process of measurement by moment balance method. (**a**) Before rotating input-micro-gear (Gear-1). (**b**) Rotated output-micro-gear (Gear-2).

**Figure 7 micromachines-15-00284-f007:**
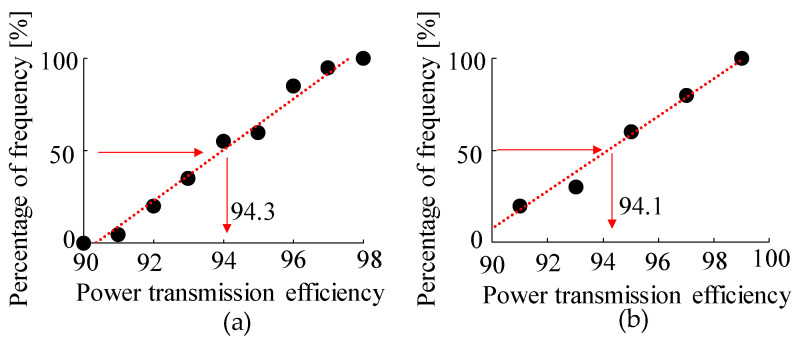
Confirmation of normality of power transmission efficiency of gears with two gears. (**a**) Macro gear. (**b**) Micro gear.

**Figure 8 micromachines-15-00284-f008:**
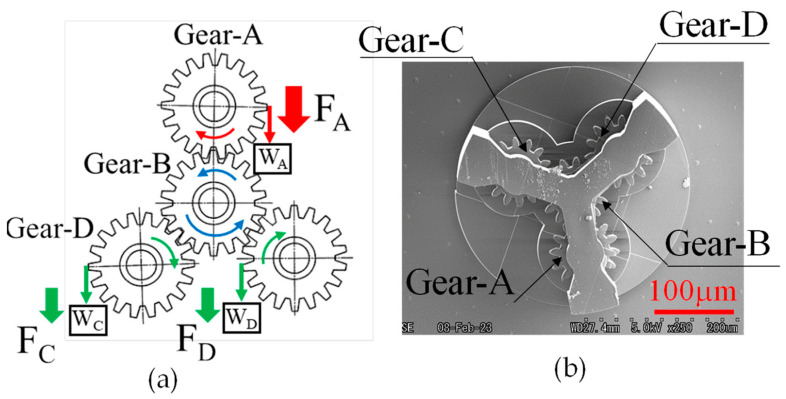
Gear train with four gears. (**a**) Schematic of the method. (**b**) SEM image of gear-train by four micro gears.

**Figure 9 micromachines-15-00284-f009:**
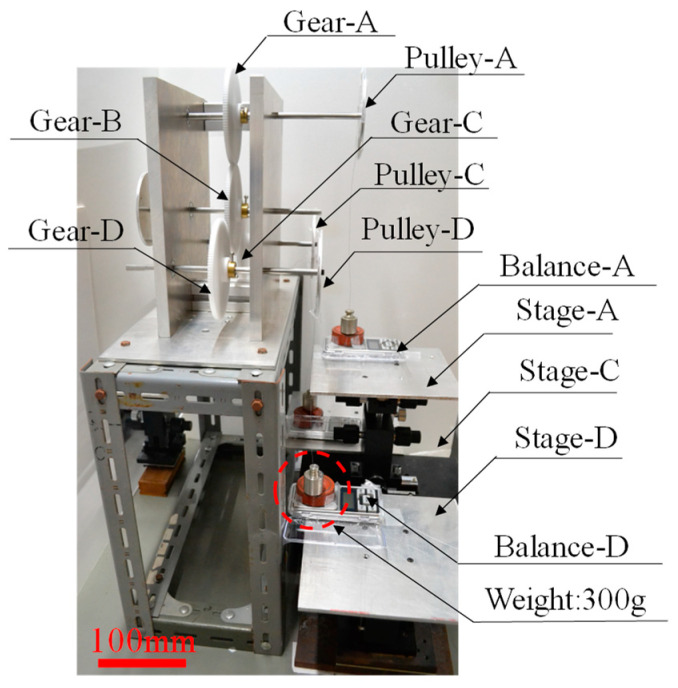
Equipment for measuring power transmission efficiency by the moment balance method in the case of four macro gears.

**Figure 10 micromachines-15-00284-f010:**
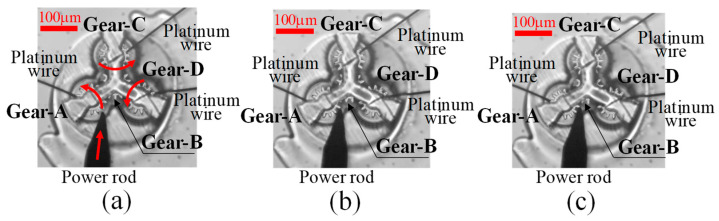
Measurement process of micro-gear train. (**a**) Setting of rotating micro-gear. (**b**) Start of rotating input-micro-gear. (**c**) Rotated output-micro-gear.

**Figure 11 micromachines-15-00284-f011:**
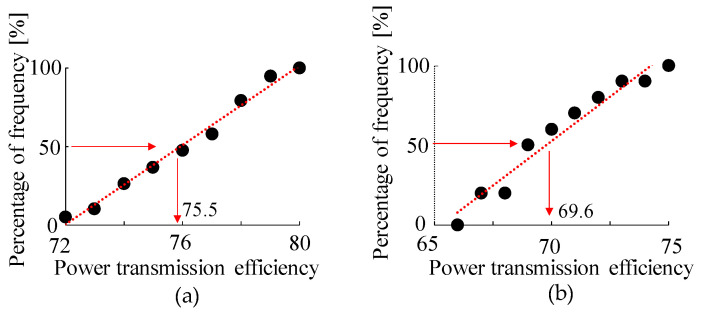
Confirmation of normality of power transmission efficiency. (**a**) Macro gear. (**b**) Micro gear.

**Table 1 micromachines-15-00284-t001:** One sample of the measured results using two gears.

	Deflection of Wire [μm]	Force [μN]	Torque [pNm]	Efficiency
Gear-1	11.43	3.06	0.129	-
Gear-2	8.78	2.87	0.121	0.937

**Table 2 micromachines-15-00284-t002:** Measured results using two gears.

	Micro-Gear Train	Macro-Gear Train
Efficiency [%]	94.1	94.3
Square sum	82.6	68.2
Unbiased dispersion	9.2	3.4
Number of measurements	10	20
*p*-value (Anderson-Darling test)	*p* = 0.573 > 0.05	*p* = 0.429 > 0.05
F0 = Vmicro/Vmacro	F0 = 2.71 < F(9, 19; 0.025) = 2.89	
*t*0	*t*0 = 0.00223 < *t*(28, 0.025) = 2.048	

**Table 3 micromachines-15-00284-t003:** One sample of the measured results using 4 micro gears.

		Deflection of Wire [μm]	Force [μN]	Torque [pNm]	Efficiency [%]
(1) Input	Gear-A	13.60	9.58	432.0	-
(2) Output	Gear-C	8.50	3.72	161.0	37.2
	Gear-D	11.20	3.56	153.0	35.5
(3)	Gear-C + Gear-D	-	-	314.0	72.7 (314.0/432.0)

**Table 4 micromachines-15-00284-t004:** Measured results using 4 micro gears.

	Micro-Gear Train	Macro-Gear Train
Efficiency [%]	69.6	75.5
Square sum	61.5	148.7
Unbiased dispersion	6.83	7.82
Number of measurements	10	20
*p*-value (Anderson-Darling test)	*p* = 0.548 > 0.05	*p* = 0.214 > 0.05
F0 = Vmicro/Vmacro	F0 =1.146 < F(19, 9; 0.025) = 2.88	
*t*0	*t*0 = 5.56 > *t*(28, 0.025) = 2.048	

## Data Availability

Data are contained within the article.

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
