# Peer review of "Consideration of Power Transmission Characteristics in a Micro-Gear Train"

_micromachines, 2024, doi:10.3390/mi15020284_

Round 1

Reviewer 1 Report

Comments and Suggestions for Authors

The author has conducted an innovative study on 2-photon polymerization (2PP) printing and its impact on spur gear transmission efficiency, offering an insightful comparison between macro and microdomains. The manuscript is promising but could benefit from a few adjustments to enhance readability and presentation.

1. In Line 99, the unit 'µm3' should be formatted with '3' as a superscript for clarity.
2. On Line 102, it would be helpful to specify the resist used - is it IP-dip or SU8? This detail could be valuable for researchers working in 2PP printing.
3. Regarding the sputtering of aluminum, is the deposition conformal? It would be beneficial to include a cross-section FIB/SEM image to verify the thickness of the deposited layer.
4. The exploration of asymmetric spur gears, particularly with different pressure angles, might offer further insights into transmission efficiency. Could this be a potential area for investigation? (The suggestion is more exploratory, I do understand this might be out of scope for this study )
5. For Figure 9, a scale bar is needed for better understanding. Also, in Figure 5, is there any slip between the platinum wire and the gear? How is consistent contact ensured throughout the experiments?
6. The arrangement of figures could be slightly reorganized to enhance the overall appearance and flow of the manuscript.
7. It would be advantageous to discuss the finite element model in more detail. Could you include some plots or schematics? Also, what contact model was used for simulating the interaction between the platinum and gear surfaces? Was it frictional or slip-based?

Finally, I recommend consolidating the conclusion into one or two paragraphs. Multiple paragraphs might lead to confusion for the reader.

Kind regards

Reviewer 2 Report

Comments and Suggestions for Authors

Issues related to MEMS devices have been developed for several years. Along with the rapid development of nanotechnology, it became possible to create such solutions in which electronics and mechanics coexist at the same time. Taking an article on this subject, I wondered what else can be revealed in this respect. First of all, the article is written by someone who really understands these issues. You can learn a lot from it. Secondly, the article contains new content related to a strict comparison of micro and macro devices, from the view of the transmitted power, depending on the degree of complexity of the device. For simple gears, there is no phenomenon related to the change of scale. For very complex devices, for micro-devices such as planetary gear, devices are not as effective as devices on a meter scale. In the case of micro-scaled devices, the effects of electrical nature come to the force and they compete with the gear mechanics. In addition, I did not notice significant editorial mistakes. The article in this form is suitable for publication.
